# Renal Medullary Carcinomas Harbor a Distinct Methylation Phenotype and Display Aberrant Methylation of Genes Related to Early Nephrogenesis

**DOI:** 10.3390/cancers14205044

**Published:** 2022-10-14

**Authors:** Victoria E. Fincke, Mateja E. Krulik, Piyush Joshi, Michael C. Frühwald, Ying-Bei Chen, Pascal D. Johann

**Affiliations:** 1Swabian Children’s Cancer Center, University Hospital Augsburg, 86156 Augsburg, Germany; 2Hopp Children’s Cancer Center (KiTZ) Heidelberg, Division of Pediatric Neurooncology, German Cancer Consortium (DKTK), German Cancer Research Center (DKFZ), 69120 Heidelberg, Germany; 3Department of Pathology and Laboratory Medicine, Memorial Sloan Kettering Cancer Center, New York, NY 10065, USA

**Keywords:** renal medullary carcinoma, DNA methylation, SMARCB1 loss

## Abstract

**Simple Summary:**

Renal medullary carcinomas (RMC) are aggressive tumors of the kidneys, characterized by a loss of SMARCB1. As the tumors, arising predominantly in young males with sickle cell trait, are very rare and no standard method for detection or treatment has been described, prognosis for these patients is poor. We generated methylation profiles of seven RMC samples and compared the hitherto unexplored methylation landscape of these tumors to other renal tumors and malignant rhabdoid tumors as well as epithelioid sarcomas, constituting two prototypically SMARCB1 aberrant entities. Based on these valuable datasets, we found that—in accordance with the previous gene expression data—RMCs separate from other SMARCB1 deficient entities. In a focused analysis of genes that are important for nephrogenesis, we particularly detected genes that govern early nephrogenesis to be hypomethylated and expressed at high levels in RMCs.

**Abstract:**

Renal medullary carcinomas (RMC) are rare aggressive tumors of the kidneys, characterized by a loss of SMARCB1. Characteristically, these tumors arise in patients with sickle cell trait or other hemoglobinopathies. Recent characterization efforts have unraveled oncogenic pathways that drive tumorigenesis. Among these, gene sets that characterize replicative stress and the innate immune response are upregulated in RMCs. Despite comprehensive genetic and transcriptomic characterizations, commonalities or differences to other SMARCB1 deficient entities so far have not been investigated. We analyzed the methylome of seven primary RMC and compared it to other SMARCB1 deficient entities such as rhabdoid tumors (RT) and epithelioid sarcomas using 850 K methylation arrays. Moreover, we evaluated the differential gene expression of RMC using RNA-sequencing in comparison to other rhabdoid tumors. In accordance with previous gene expression data, we found that RMCs separate from other SMARCB1 deficient entities, pointing to a potentially different cell of origin and a role of additional genetic aberrations that may drive tumorigenesis and thus alter the methylome when compared to rhabdoid tumors. In a focused analysis of genes that are important for nephrogenesis, we particularly detected genes that govern early nephrogenesis such as FOXI1 to be hypomethylated and expressed at high levels in RMC. Overall, our analyses underscore the fact that RMCs represent a separate entity with limited similarities to rhabdoid tumors, warranting specific treatment tailored to the aggressiveness of the disease.

## 1. Introduction

Renal medullary carcinoma (RMC) is a rare tumor of the kidney, that mainly arises in young males of African descent with sickle cell trait or, rarely, other sickle cell hemoglobinpathies. The pathogenesis of these tumors is incompletely understood, but the hypoxic milieu in the kidney may confer a particular susceptibility for genomic instability, loss of the SMARCB1 (also known as INI1) protein and thus tumor formation [1]. Despite its rarity, it is the third most frequent renal cancer in adolescents [2]. Due to its aggressiveness, RMC has a very poor prognosis for patients and metastases are common at the time of presentation. As patient numbers are low, no standard of care has been defined. Treatment consists of comprehensive multimodal therapy including nephrectomy, chemo-, and radiotherapy. Nonetheless, the mean overall survival is only 6–8 months [3] and fewer than 5% of patients survive for more than 36 months [4]. Alternative treatments with small molecules (e.g., EZH2 inhibitors) and immunotherapy have been investigated as options for treatment but remain without resounding success [5]. Reliable methods for the early detection of these rare tumors have not been described [6]. Except for the strong clinical link to sickle cell trait, the underlying biology of the RMC pathogenesis or its cell-of-origin is only poorly understood.

RMCs are characterized by a loss of the tumor suppressor SMARCB1 (SWI/SNF-related matrix-associated actin-dependent regulator of chromatin subfamily B member 1) protein expression, a core member of the SWI/SNF (SWItch/Sucrose Non-Fermentable) complex [7,8]. This complex is an ATPase-dependent multisubunit complex involved in transcriptional regulation and chromatin remodeling [6]. In RMC, the biallelic inactivation of SMARCB1 occurs largely either via concurrent hemizygous loss and translocation disrupting SMARCB1 or by homozygous deletion [9,10,11]. The hypertonicity and hypoxia within the renal medulla and the location of *SMARCB1* at 22q11.2, a known hotspot for de novo deletions and translocations that occur under hypoxic stress, are important factors in the progression of RMCs [12]. A recent study by Vokshi et al. [13] showed an activation of ferroptosis resistance pathways in RMC cells, which helps to link the RMC oncogenic process to its association with the sickle cell trait. The increased iron concentration in the extracellular space caused by sickled red blood cells (RBCs) leads to selective pressure for ferroptosis resistant cells, which seem to be cells harboring a SMARCB1 loss [13].

The absence of SMARCB1 in mouse models leads to early embryonic lethality, and mice with conditional SMARCB1 loss develop aggressive tumors at a median age of 11 weeks [14]. Other SMARCB1 deficient cancers include epithelioid sarcomas (ES) and malignant rhabdoid tumors (MRTs), called atypical teratoid rhabdoid tumors (ATRTs) when developing in the central nervous system (CNS). Both occur predominantly in young children. These tumors, despite their common SMARCB1 loss, have widely varying clinical traits, which may at least in part be explained by a different epigenetic rewiring of the respective cells of origin. For ATRT, three distinct subgroups have been identified and revealed varying degrees of hypermethylation [15]. Methylation analysis in other kidney cancers has shown widespread promoter hypermethylation [16], but methylation data for RMCs are lacking. Extracranial malignant rhabdoid tumors (eMRTs) and RMCs were shown to have a rather low mutational burden, what indicates that SMARCB1 deficiency is the driver of these highly malignant tumors [17,18,19]. Based on their similar background of SMARCB1 loss, as has already been stated, RMCs and rhabdoid tumors of the kidney (RTK) share a similar gene expression signature, which is distinct from other renal tumors [10].

A recent comprehensive tumor characterization of renal medullary carcinomas by Msaouel et al. [5] at the genetic and transcriptomic level has characterized the genome and unraveled deregulated pathways that may drive RMC tumorigenesis. Except for occasional aberrations in the gene *SETD2*, no recurrent SNVs or INDELs (insertion/deletions) beyond *SMARCB1* have been identified. However, recurrent amplifications of *NOTCH2* point to a potential overactivation of this pathway. Moreover, an overactivation of *MYC* gene expression and its targets has been found to contribute to replicative stress in these tumors. In this study, transcriptomic analysis also demonstrated clear differences between rhabdoid tumors and RMCs. On one hand, this points toward a different underlying biology of these entities; on the other hand, many similarities between various SMARCB1 and/or different SMARCA4 tumors such as ATRT and small cell carcinomas of the ovary hypercalcemic type (SCCOHT) [20], ATRT and cribiform neuroepithelial tumors (CRINET) [21] as well as ATRT and SMARCB1-deficient chordomas [22] have been described.

To our knowledge, epigenetics has so far not been studied in RMCs. We therefore generated the methylation profiles of seven RMC samples, three of which had been previously published in Jia et al. [9], and compared the hitherto unexplored methylation landscape of these tumors to other renal tumors (papillary renal cell carcinoma, clear cell carcinoma) and malignant rhabdoid tumors as well as epithelioid sarcomas, constituting two prototypically SMARCB1 aberrant entities. We also included the methylation of Wilms tumors into the analyses as they represent approximately 95% of pediatric kidney tumors [23,24].

## 2. Material and Methods

### 2.1. Tumor Samples

For all of the analyses, we used unpublished and published datasets. RMC samples were provided by Dr. Ying-Bei Chen (Department of Pathology, Memorial Sloan Kettering Cancer Center, New York, PMID30980040). Three of the seven samples have been previously published, as detailed in Appendix A. Loss of SMARCB1 in the RMCs was confirmed by IHC (Figure 1E exemplarily). For comparison, seven kidney renal clear cell carcinomas (KIRC) and six papillary cell renal carcinomas (KIPR) and samples were chosen; IDAT files were downloaded from the TCGA data portal (https://portal.gdc.cancer.gov/, (accessed on 5 January 2019)). For ATRT, all reference methylation datasets/samples were derived from the ATRT consensus paper [25], and the SCCOHT samples are published in Fahiminyia et al. [20]. The eMRT and RTK samples were previously published by Chun et al. [26]. Data for Wilms tumors and epithelioid sarcomas (ES) were generated as comparators for this study.

### 2.2. DNA Methylation Profiling

DNA was isolated from formalin-fixed paraffin-embedded tumor samples, purification, and bisulfite conversion were performed as described before [27]. Either the HumanMethylation450 BeadChip array or the Infinium MethylationEPIC BeadChip Kit were used in order to quantify the methylation levels of 450,000 or 850,000 CpG sites per sample.

### 2.3. Bioinformatics

IDAT files were processed as previously described [28]. In detail, the R package minfi (V1.32.0) was used to load the data into R. For t-distributed stochastic neighbor embedding (tSNE) analysis, the Rtsne (v.0.15) R package was used. The heatmap was built utilizing the R package pheatmap (v.1.0.12) based on the 5000 most variable CpG sites identified by standard deviation. Red represents hypermethylation and blue represents hypomethylation. Violin plots were generated using the R package vioplot (v.0.3.7).

For analysis of differential methylation, we calculated an aggregated methylation value per gene. Therefore, we averaged the beta values of each CpG site for every gene that is represented on the 450 K array. To this end, all CpG values, overlapping with the gene body or transcription start site, were considered.

The resulting methylation values per gene were subjected to testing for differential methylation by using the Student’s t-test with Welch correction.

Gene set enrichment analysis was performed using Consensus Path DB provided by the Max Planck Institute for Molecular Genetics (www.cpdb.molgen.mpg.de, (accessed on 1 February 2022)), analyzing enriched sets based on Gene Ontology. Copy-number alterations analyzed from methylation array data were performed utilizing the conumee Bioconductor package (1.28.0) and chromosomal gains or losses were manually examined.

## 3. Results

Tumor methylation profiling revealed a close proximity of RMCs (*n* = 7) to epithelioid sarcomas (*n* = 7) with respect to their methylation profiles: t-distributed stochastic neighbor joining (t-SNE) analysis showed RMCs in close proximity to ES. In line with this, these entities formed a separate cluster in the unsupervised hierarchical clustering analysis of the 5000 most variable CpG sites. Remarkably, the kidney entities analyzed, KIPR (*n* = 6) and KIRC (*n* = 7), and the pediatric entity of Wilms tumors (*n* = 13), differed substantially in the methylation profiling. It is known that the genome of ATRT is largely devoid of chromosomal aberrations apart from the characteristic loss of SMARCB1 in 22q11 [29]. To assess whether the same is true for RMCs, the copy number profiles derived from the methylation array data were examined for alterations. Apart from the expected loss of 22q (100%, *n* = 6), the loss of 4q, 8q (both 33%), and 15q (50%) was observed. Non-recurrent losses affecting 9q, 13q, 14q, 16q, and 17q were detected in single cases of the cohort.

It is noteworthy that SMARCB1 aberrations also occur in non-RMC and non-rhabdoid kidney tumors. To delineate if there are epigenetic and thus potentially biologic similarities between this subset of renal tumors, we investigated them in a separate analysis (Appendix A).

When comparing the RMC methylation profile to further renal tumor entities (e.g., papillary renal cell carcinomas (PRCC) type 1 and type 2, unclassified PRCC, and chr22_loss) by unsupervised hierarchical clustering, the close proximity between RMCs and ES remained unchanged (Appendix A). This corroborates the study from Msaouel et al. [5], which revealed clear differences between the RMC and PRCC gene expression levels in kidney nephron sites.

Aiming to gather additional insights into the regulation of gene expression by methylation, we calculated the averaged methylation values per gene for all genes that were covered by the methylation array.

Genes involved in nephrogenesis were found to be differentially methylated in RMCs compared to KIPRs and KIRCs (Figure 2D). Genes regulating the posterior intermediate mesenchyme (PIM), the metanephric mesenchyme (MM), and the cap mesenchyme were predominantly hypomethylated in RMCs, whereas genes involved in the progression from the renal vesicle (RV) to the comma-shaped body (CSB) and later the S-shaped body (SSB) were hypermethylated (Figure 2A). Ureteric bud induction involved genes showing changes in methylation in RMCs in both directions. An example of a hypomethylated gene in RMCs is *LIM homeobox 1* (*LHX1*), encoding a transcription factor that has particular importance in the formation of the intermediate mesoderm and the nephric mesenchyme. The methylation of this gene was significantly elevated in the RMCs (0.977) compared to KIPR (0.95, *p* = 0.0012) and KIRC (0.955, *p* = 0.0034) samples and when compared to the different ATRT subgroups (TYR= 0.916, *p* = 6.4 × 10^−8^; SHH = 0.926, *p* = 1.5 × 10^−6^; MYC = 0.9, *p* = 6.7 × 10^−6^). An example of a hypomethylated gene in RMCs is *Forkhead box I1,* also known as *FOXI1*, a gene encoding a transcriptional activator required for kidney development. Overall methylation of *FOXI1* in RMCs was 0.232, which was lower compared to the methylation of the same gene in KIPR (0.582, *p* = 0.00016), KIRC (0.576, *p* = 7.2 × 10^−5^), ATRT-TYR (0.34, *p* = 0.088), ATRT-SHH (0.422, *p* = 0.0066), and ATRT-MYC (0.347, *p* = 0.078) (Figure 2B,C). The methylation pattern of genes involved in nephrogenesis is quite similar in RMCs and Wilms tumors, although analyses of the most differentially methylated CpG sites showed clear differences and led to distinct clustering (Figure 1). Only four of the 27 analyzed genes related to kidney development were significantly differentially methylated in Wilms tumors; *CDH1* is hypomethylated in Wilms tumors (mean = 0.357) compared to RMCs (mean = 0.559, *p* = 0.039) as well as *KDR* (*p* = 0.028), *FGFR2* (*p* = 0.007), while *GATA3* (mean = 0.74) is hypermethylated compared to RMCs (mean = 0.566, *p* = 0.0005) (Appendix A). To corroborate our methylation analysis with transcriptomic data, we generated RNA-Seq data from the FFPE material of the RMC samples. Unfortunately, as comparators, we only had rhabdoid tumors available as a comparison with RNA-Seq data from other kidney tumors (KIPR, KIRC), which was generated from fresh frozen material that would suffer from extensive batch effects.

In order to avoid these, we only compared datasets for RMCs, TYR, MYC, and SHH. For the example of *LHX1*, which is hypermethylated in RMCs, it shows that the RNA expression of *LHX1* was lower in RMCs (mean = 5.849) than in TYR (mean = 6.372, *p* = 0.508), MYC (mean = 6.903, *p* = 0.19), and SHH (mean = 6.292, *p* = 0.525).

Hypoxia plays a role in the development of RMCs as the environment is characterized by extreme hypoxia and hypertrophy of the renal medulla. It has been previously shown that hypoxia induced genes are expressed at a higher level in RMCs compared to adjacent normal tissue [5]. Our findings support the results from Msaouel et al. that showed these genes to be hypomethylated in our analysis. Other genes involved in epithelial mesenchymal transition (EMT) were hypomethylated in RMCs compared to KIPRs and KIRCs (Figure 3). *CEACAM6*, encoding for the cell adhesion protein CEACAM6, is hypomethylated in RMCs. CEACMA6 is known to exert a pro-invasive role in several carcinomas via regulation of matrix metalloprotease-9 (MMP9) [30], which is also hypomethylated. It has previously been shown that the expression of both genes is upregulated in RMCs [5], which corroborates our results. Additionally, our findings are in line with the study by Vokshi et al., which showed clusters of cells harboring an EMT signature in single cell transcriptome analyses of a RMC patient sample as well as an enrichment of EMT and hypoxia in two independently generated RMC cell lines [13]. Although RMCs and ES show similarities in global methylation, as seen in hierarchical clustering (Figure 1B), the methylation of genes involved in nephrogenesis showed clear differences. For example, *HOXD11* was significantly different methylated in ES (mean = 0.019) compared to RMCs (mean = 0.042, *p* = 0.029), which is also true for *GREM1* (ES mean = 0.833, RMC mean = 0.668, *p* = 0.48), while other genes show similar methylation values per gene such as *IL6* (ES mean = 0.959, *p* = 0.0056) (Appendix A).

As RMCs show the invasion of inflammatory cells such as neutrophils [31], we evaluated the methylation aberrations of neutrophil marker genes. Five of the nine neutrophil marker genes were differently methylated in RMCs compared to KIRC and KIPR. Lin, CD11b, CD33, and CD44 were hypomethylated, while CD55 was hypermethylated in RMCs compared to the other kidney tumors. Myeloperoxidase, CD15, CD45, and integrin alpha 4 showed no differences in methylation (see Appendix A).

## 4. Discussion

In this study, we analyzed the methylome of seven RMC samples and compared them to the methylation data of other SMARCB1 deficient tumor entities. We speculated that the investigation of the methylome in these tumors may help to achieve a more precise molecular classification, which is important as little is known on the epigenome and transcriptome of these tumors.

We performed DNA methylation and RNA sequencing analysis to examine the RMCs on a molecular level. RMCs showed the closest proximity to epithelioid sarcomas, but distinct from the group of rhabdoid tumors. This similarity is remarkable, given the different organ systems that these tumors arise in and thus the different cells of origin that may give rise to these. Beyond the epigenetic level, there were other similarities between RMCs and ES that may merit further investigation: both ES and RMCs were accompanied by further cytogenetic abnormalities that may facilitate tumorigenesis. While the impact of these aberrations remains unclear, it may well be supposed that SMARCB1 might not be the only oncogenic driver of RMCs, just as it is already presumed for epithelioid sarcomas [32].

Our data corroborated the findings of Msaouel et al., which point to clear epigenetic differences between rhabdoid tumors and renal medullary carcinomas. It is thus unclear whether common epigenetic targets [33], which have been exploited to treat rhabdoid tumor patients, may have similar, beneficial effects as in RT. To more precisely compare the drug target expression of these tumors, further comparative studies are needed that may also involve other data dimensions such as the proteome. Moreover, cell line models that faithfully recapitulate the biology of RMCs are needed and have recently been established [34,35].

With respect to the aberrant epigenetic regulation that may underlie the associations between epithelioid sarcomas and renal medullary carcinomas, these aforementioned functional studies may elucidate the similarity between ES and RMCs. Methylation of genes involved in nephrogenesis showed no obvious resemblance in RMCs and ES, therefore, the biology underlying the analogy in the overall methylation of these entities needs to be investigated further.

Our analysis of differentially methylated genes by comparing RMCs with other kidney tumors predominantly revealed genes of early nephrogenesis to be hypomethylated in RMCs: *EYA1*, for instance, a transcription factor that has been shown to be a key initiator of MM development [36], is hypomethylated compared to KIPR and KIRCs. Similarly, *PAX2*, another TF implicated in epithelialization of the mesenchyme, is hypomethylated in RMC. Unfortunately, our study lacks methylation data from the developing kidney which, to our knowledge, is not available for the EPIC methylation array platform. However, the pattern identified here when comparing RMCs and other kidney tumors may point to a developmental arrest of RMCs at relatively early stages of kidney development. Along a similar line of evidence, the EMT genes *OCD1* and *CEACAM6* were found to be hypomethylated in RMCs—these are key protagonists in controlling the EMT switch, but also important players in the formation of metastases [37] in different tumors.

The similarity of methylation in RMCs and Wilms tumors, a pediatric malignancy with features of stagnant kidney development [38,39,40], needs to be assessed further in analyses with more omics data in order to reveal analogies in the biology of the entities.

Taken together, our data underline the molecular distinction between eMRT and RMCs but also show differences between RMCs to common types of RCC (KIRP and KIRC). Our analysis also suggests that the identification of the dysregulated methylation of early kidney development represents an important aspect of RMC biology that may inform subsequent preclinical and clinical studies.

Due to the rarity of these tumors, our study was clearly limited by the small number of samples included. A further, thorough characterization of these tumors is thus needed that also includes the comparability of the published cell line models [34] with their respective primary counterparts.

## 5. Conclusions

The methylome RMCs showed the closest proximity to epithelioid sarcomas, but is distinct from the group of rhabdoid tumors. Our analysis of differentially methylated genes by comparing RMCs with other kidney tumors predominantly revealed genes of early nephrogenesis to be hypomethylated in RMCs and showed distinct methylation patterns of genes involved in EMT. The investigation of the methylome of RMCs may help to archive a more precise molecular classification, but further investigations are needed in order to fully characterize these tumors and -ultimately- improve treatment options. 

## Figures and Tables

**Figure 1 cancers-14-05044-f001:**
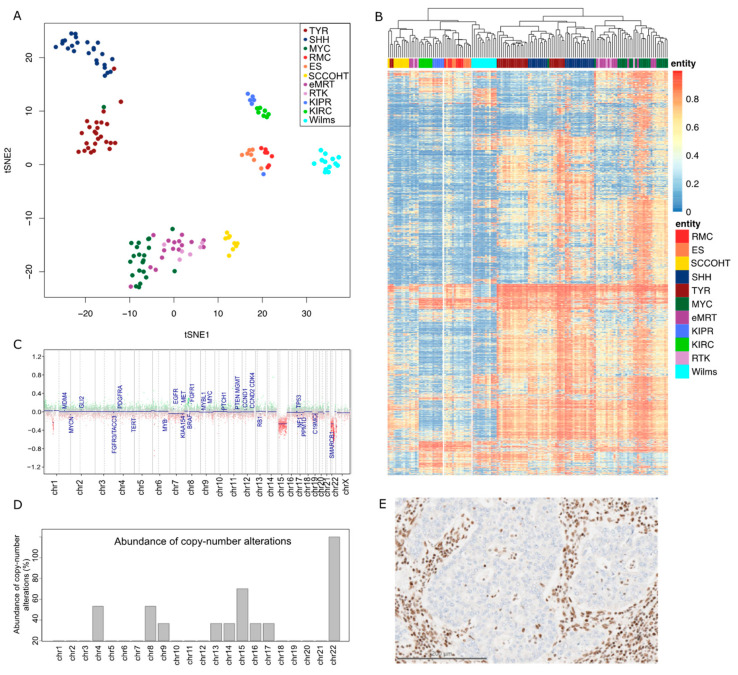
Epigenetic signature distinguishing RMC from other tumor entities. (**A**) Unsupervised tSNE analysis based on the DNA methylation profiles of RMCs (*n* = 7), SCCOHTs (*n* = 9), ATRT-TYR (*n* = 25), ATRT-SHH (*n* = 25), ATRT-MYC (*n* = 22), epithelioid sarcomas (*n* = 7), eMRTs (*n* = 16), RTK (*n* = 6), KIPRs (*n* = 6), KIRCs (*n* = 7), and Wilms tumors (*n* = 13). (**B**) Unsupervised hierarchical clustering based on the 5000 most variably methylated CpG sites. (**C**) Example of a copy number profile from a female RMC patient with a characteristic chromosomal loss on chr22 and an additional loss on chr15. (**D**) Abundance of copy number alterations in percent in the complete cohort. (**E**) Expression of SMARCB1 is lost in tumor cells, while normal nuclear staining of SMARCB1 is retained in endothelial and stromal cells and the benign renal tubules.

**Figure 2 cancers-14-05044-f002:**
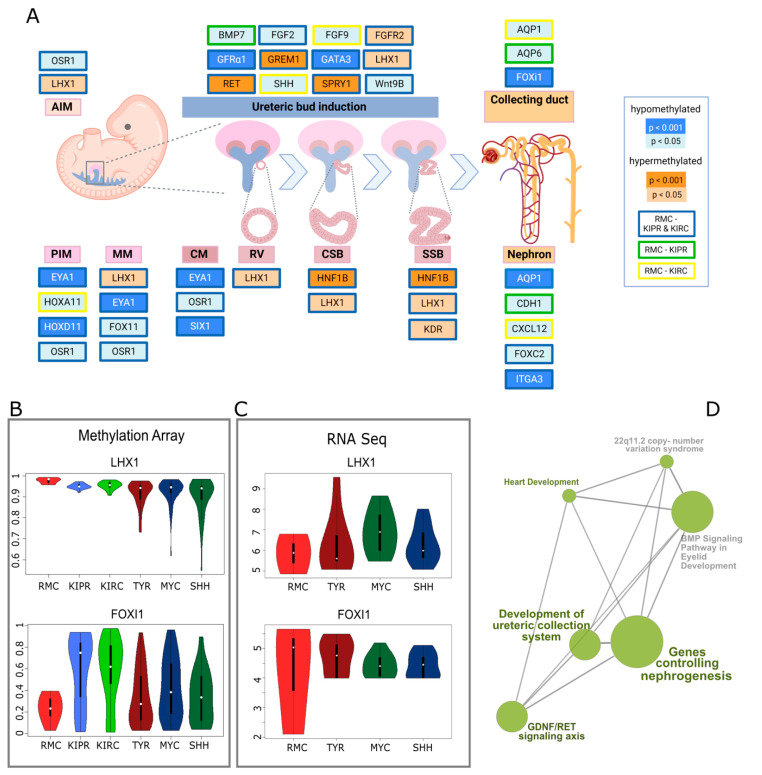
Altered DNA methylation of genes involved in nephrogenesis. (**A**) Pathway diagram of genes involved in the different stages of human nephrogenesis representing significant differences in methylation between RMC and two other kidney carcinomas—KIPR and KIRC. PIM, posterior intermediate mesenchyme; AIM, anterior intermediate mesenchyme; CM, cap mesenchyme; RV, renal vesicle; CSB, comma-shaped body; SSB, S-shaped body; created with BioRender. (**B**) Violin plots displaying the DNA methylation of the genes *LHX1* and *FOXI1* for RMC, KIPR, KIRC, ATRT-TYR, ATRT-SHH, and ATRT-MYC. (**C**) Violin plots showing the RNA expression of *LHX1* and FOXI1 for RMC, ATRT-TYR, ATRT-MYC, and ATRT-SHH. (**D**) Gene sets analyzed based on ontology with aberrant methylation in RMCs and their connections computed with consensus path DB.

**Figure 3 cancers-14-05044-f003:**
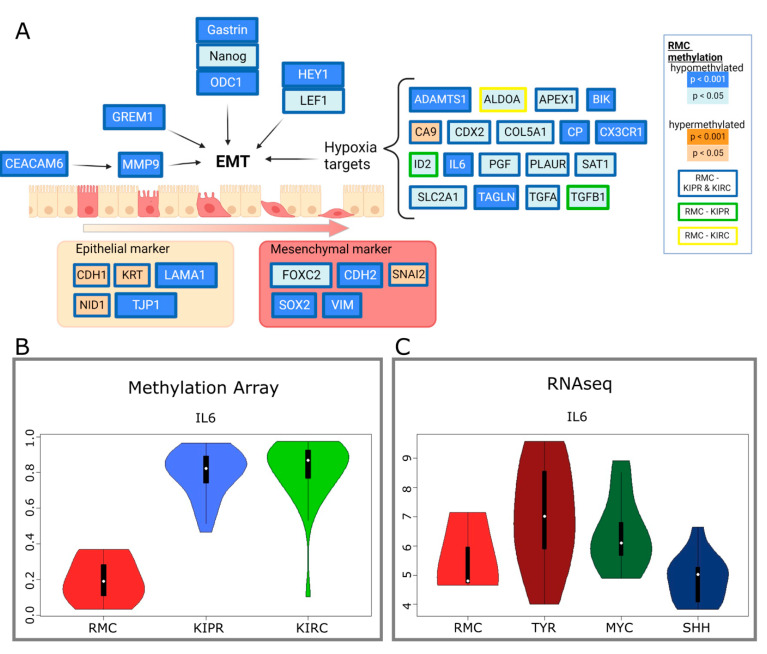
Altered methylation of genes involved in epithelial mesenchymal transition (EMT). (**A**) Diagram of genes involved in EMT including hypoxia-induced genes, epithelial, and mesenchymal markers, showing their DNA methylation compared to KIPR and KIRC. Coloring indicates the *p*-value level as indicated in the graphics. Created with BioRender. (**B**) Violin plots displaying the methylation of IL6 in RMC, KIPR, and KIRC. (**C**) RNA-Seq data presented in violin plots showing the RNA expression of IL6 in RMC, ATRT-TYR, ATRT-MYC, and ATRT-SHH.

## Data Availability

The new data presented in this study will be available at NCBI GEO via the accession number: GSE211942.

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
