# Peer review of "Renal Medullary Carcinomas Harbor a Distinct Methylation Phenotype and Display Aberrant Methylation of Genes Related to Early Nephrogenesis"

_cancers, 2022, doi:10.3390/cancers14205044_

Round 1

Reviewer 1 Report

Fincke et al’s manuscript “RMC harbor a distinct methylation phenotype and display aberrant methylation of genes related to early nephrogenesis” provides an overview of methylation studies performed on 7 patients with RMC.

The authors compare the results with adult KIRC and KIPR samples and compared this to ATRG, SCCOHT eMRT, and ES profiles.

Overall, this is an important manuscript for the field but requires further analyses to better characterize the methylation landscape in RMC.

Major

1.       Prior studies (which ideally would also be cited to be complete) have shown that RMC & malignant rhabdoid tumors of the kidney may share similarities. This manuscript briefly touches upon this and it would be helpful to expound more and perform some additional analyses to this effect. To determine if methylation status is similar, it seems like the authors have a dataset already but just in case, one could use the one derived from TARGET: https://www.cell.com/cancer-cell/fulltext/S1535-6108(16)30043-5

2.       Given the discussion of nephron progenitors, prior work in Wilms Tumor also would be helpful to better understand if these insights – similar to #1, the discussion of lines 180-193 are very much related to WT / MRT biology so further comparisons / analyses would be helpful. Further incorporation into Figures 2A and 3A would be helpful.

3.       Can the authors comment on the loss of 22q – was this a deep deletion? Can the authors provide additional details if these had a balanced translocation or if these were from large biallelic deletions? Supp 1 suggests that there were no deletions in RMC003, 004 & 006? Can the authors comment how SMARCB1 is lost in these cases or show IHC showing loss of SMARCB1?

4.       For figure 1b, can the authors comment / add information on the clustering by the y-axis (as shown in the supp fig)? Given the findings, particularly developmental biology related clustering vs lineage?

5.       Given the emphasis on the similarities between ES and RMC, would be helpful to have more analyses behind that in the results section.

5.       There needs to be discussion on how one corrected for batch effects given that some samples came from FFPE vs fresh frozen.

Minor

1.       Methods – please clarify where the eMRT and other subpopulation of samples data came from as this is not clear in the tumor samples used.

2.       Throughout the manuscript, the authors state that RMC occurs by a “-mostly-“ biallelic loss of SMARCB1 – recent papers suggest that this is actually the case (e.g., balanced translocations which were seen in the majority of cases in the COG experience) – would recommend clarifying (as noted above in #3)

3.       Figure labels are difficult to read when printed out. Also, unclear why the figure backgrounds are highlighted differently for methylation vs RNAseq

4.       Of note, several groups have established cell line models of RMC (in lines 265 and 288) so may be worthwhile to cite those as well.

Reviewer 2 Report

This manuscript is well written to report methylome of renal medullary carcinoma, which is intimately with sickle cell anemia and an aggressive but rare type of renal cell carcinoma.

Round 2

Reviewer 1 Report

Just minor edits to suggest. Otherwise appreciate the authors for their work.

-Correspondence - missing a contact email / address

-Abstract - correct FOXI to FOXI1

For the figures,

-text in Fig 1A & C is a bit difficult read - consider enlarging the text

-Fig 1B - figure seems cut off on the Right side (on the printed version)

-Fig 2 & 3 - the black text on dark blue background is difficult to read

-text in 2B & C are just very small again
